# Pro-Environmental Behavior: Examining the Role of Ecological Value Cognition, Environmental Attitude, and Place Attachment among Rural Farmers in China

**DOI:** 10.3390/ijerph192417011

**Published:** 2022-12-18

**Authors:** Lin Meng, Wentao Si

**Affiliations:** School of Public Administration, Shandong Normal University, Jinan 250014, China

**Keywords:** ecological value cognition, pro-environmental behavior, environmental attitude, place attachment, Jinan, China

## Abstract

Studies on the factors that influence farmers’ pro-environmental behavior could promote environmental management in rural areas. Jinan of China was selected as the case study area in this study. A structural equation model and multiple hierarchical regression analysis were applied to analyze the influence mechanism of ecological value cognition on pro-environmental behavior. Environmental attitudes were set as the mediating variable and place attachment was selected as the moderating variable. The results showed that (1) ecological value cognition exhibited a positive influence on pro-environmental behavior in both direct and indirect ways. The indirect influence was mediated by environmental attitude. (2) Place identity and place dependence showed a positive direct influence on pro-environmental behavior. (3) It is suggested that in order to improve pro-environmental behavior, enhancing ecological value cognition, cultivating farmers’ positive environmental attitude, increasing farmers’ place attachment, and releasing reward and punishment measures are good strategies. The findings in this study are important to the improvement of the rural ecological environment and the quality of life of farmers. Meanwhile, the findings shed light on the construction process of ecological civilization and the improvement of public welfare.

## 1. Introduction

Farmers’ pro-environmental behavior is behavior that is good for environmental protection when farmers encounter ecological problems [1,2]. Pro-environmental behavior can improve environmental management in rural areas [3]. Chinese rural areas are vast in territory, large in population, and most closely related to natural ecology [4]. In order to improve the rural environment and spread the concept of ecological civilization, China began to carry out a series of construction projects such as “the most beautiful countryside in China” and “civilized ecological village” in 2002, which received positive responses and support from all sectors of society. However, compared with cities, economic development in rural areas is slow. The natural ecological environment in rural areas is complex and diverse. The government’s investment in environmental protection facilities is relatively insufficient compared with that in urban areas [5]. The actual situation of rural ecological environment protection is still unsatisfactory. In total, 1,310,000 tons of agricultural chemicals and 52,507,000 tons of chemical fertilizer were used in China in 2020 [6]. Water resources, soil, and agricultural products were greatly damaged by this kind of agricultural production manner, which blocked the sustainable development of the agricultural economy in China. Central Document No. 1 in China points out that it is important to control non-point pollution in agriculture. However, due to the dispersibility and crypticity of internal pollution, as well as the deficiency of labor and funding, “government failures” still exist in rural environmental management [7,8]. New institutional economics consider that induced institutional changes are more effective than imposed institutional changes [9]. Thus, the protection of the rural ecological environment requires the participation of the whole society, and more importantly, it requires rural residents to play the primary role. Compared with imposed institutional changes that the government controls, techniques that drive farmers to perform pro-environmental behavior positively are better for solving environmental problems.

The detection of factors that influence farmers’ pro-environmental behavior has become a hot topic in policy making and scientific research. According to past studies, value cognition is an important factor that influences farmers’ pro-environmental behavior. Under the assumption that farmers are rational economic individuals, farmers’ behavior is the result of weighing the pros and cons [10]. The cognition of physical capital [11,12], human capital [13], and social capital [14] influences farmers’ pro-environmental behavior. What is more, essentially speaking, the process of farmers’ pro-environmental behavior improvement is exactly the same process of improving the ecological value in rural areas because farmers’ welfare will be improved in the process [15,16]. Therefore, whether farmers perform pro-environmental behavior is determined by their ecological value cognition. In previous studies, scientists noticed the positive effect of ecological value cognition on pro-environmental behavior and proved that ecological value cognition has a positive effect on pro-environmental behavior such as endangered species protection [16], water resource protection [17], returning grain plots to forestry [18], carbon sequestration [19,20,21], and the application of agricultural technology [22]. However, ecological value cognition is a complex, abstract conception. The definition is vague and is always mixed with environmental consciousness and environmental concern [23,24]. Environmental consciousness refers to the individual’s beliefs and emotional and behavioral tendencies regarding the concepts and events related to environmental protection [25]. Environmental concern refers to the extent that farmers care about their surrounding environment. There are obvious differences between environmental consciousness, environmental concern, and value cognition [26]. Therefore, it is necessary to propose a clear definition of ecological value cognition and address the influence mechanism of ecological value cognition on pro-environmental behavior.

In addition, existing research also analyzed the main factors that affect farmers’ pro-environmental behavior regarding demographic characteristics, psychological factors, and other aspects. First, demographic characteristics research mainly focused on the relationship between demographic variables (such as gender, age, education level, and income level) and pro-environmental behavior [27,28,29]. However, such studies did not reach a consistent conclusion. Therefore, the consistency of research results on the influence of demographic factors (such as gender, age, education level, and income level) on farmers’ pro-environmental behavior is relatively limited. Second, concerning the impact of psychological factors on pro-environmental behaviors, studies found that attitude, place attachment, subjective norms, moral norms, etc., had different degrees of impact on pro-environmental behavior. Related studies were mainly based on the Theory of Planned Behavior (TPB) [1], the Norm Activation Model (NAM) [30], and the Value–Belief–norm Theory (VBN) [1]. Psychological factors were proven to have an excellent explanatory ability in pro-environmental behavior analysis, of which environmental attitude was always defined as a key factor in addressing the complex internal process and psychological mechanisms in farmers’ pro-environmental behavior [31]. The place attachment that formed due to the special rural culture in China has an influence on both environmental attitudes and pro-environmental behaviors [32]. However, previous studies have always considered place attachment as one antecedent variable, neglecting discussion on its moderating effect.

To summarize, it is of great significance to study the relationship between farmers’ ecological value cognition, environmental attitudes, place attachment, and pro-environmental behavior. On the one hand, there are few research studies on the relationship between farmers’ ecological value cognition, environmental attitudes, place attachment, and pro-environmental behavior, and most of the existing research has focused on the relationship between environmental attitudes and pro-environmental behavior. Taking ecological value cognition as the core variable, exploring the relationship between ecological value cognition, environmental attitudes, place attachment, and pro-environmental behavior can effectively help scientists to find the root of environmental problems and fundamentally change environmental attitudes and behaviors. On the other hand, in the face of increasingly serious rural environmental problems, more and more governments and individuals have begun to make contributions to environmental protection. They have tried to use a variety of methods to arouse farmers’ awareness of environmental protection and reduce environmental damage. In this process, people have become more and more aware of the relationship between man and nature behind these problems, as well as the relationship between cognition, psychology, and behavior. Therefore, it is not enough to rely only on scientific, technological, economic, and legal means to solve environmental problems. Every human practice is conducted under the guidance of one certain type of cognition. The deepest guiding ideology is cognition. Thus, the causes of human damage to the natural environment should first be found in ecological value cognition. Correct ecological value cognition is the fundamental countermeasure to solving environmental ecological problems and realizing sustainable development. Therefore, an in-depth study on the relationship between farmers’ ecological value cognition and pro-environmental behavior conforms to the needs for environmental protection and the psychological needs of the public.

In this study, Jinan was selected as the study area, the environmental attitude was set as the mediating variable, and place attachment was selected as the moderating variable to study the influence mechanism of ecological value cognition on pro-environmental behavior. Three questions were posed: (1) Does ecological value cognition have a direct influence on pro-environmental behavior? (2) Does environmental attitude play a mediating role in the influence of ecological value cognition on pro-environmental behavior? (3) Does place attachment regulate the influence of environmental attitude on pro-environmental behavior? The findings in this study are important for the improvement of the rural ecological environment and the quality of life of farmers. Meanwhile, the findings shed light on the construction process of ecological civilization and the improvement of public welfare.

## 2. Literature Review and Research Hypothesis

### 2.1. The Definition of Ecological Value Cognition

Ecological value cognition is farmers’ cognition of service value provided by the ecosystem. The concepts of the labor theory of value, traditional economic value theory, and environmentalism axiology were introduced to ecological value cognition during its conceptual development. According to the labor theory of value, human work created the use value of the ecosystem [33,34]. Traditional economists hold the view that ecological value is generated in the process of market trading where the use value of an ecosystem is transferred between the supplier and the buyer [35]. Environmentalists think that ecosystems have internal value, of which their existence is not determined by the subjective willingness and feelings of human beings [36]. To summarize, ecological value is the service value of the ecosystem [22,24].

The System of Environmental Economic Accounting (SEEA) is a worldwide standard in environmental economic accounting [37]. SEEA considers ecosystem services as the contribution or benefit of the ecosystem to the economic system. According to the function of the ecosystem and the manner in which humans use it, SEEA divides ecosystem services into regulating services, provisioning services, and cultural services [38]. Regulating services refer to the ability of the ecosystem to regulate ecological processes such as climate, hydrology, and the biochemical cycle. Provisioning services are the material and energy that the ecosystem provides to the economic system. Cultural services represent the knowledge, pleasure, and satisfaction that humans obtain from the ecosystem. Provisioning services could be understood as trade in the market, with the attribute of personal items. The key point of its value calculation is the production cost and producer surplus, which reflect the concept of exchange value [39]. Regulating services and cultural services have the attributes of public goods. The key point of their value calculation is consumer surplus, which reflects the concept of welfare economic value [37]. To summarize, ecological value cognition refers to people’s understanding of the natural environmental conditions and utility values that are formed and maintained by the ecosystem and ecological process on which human beings rely for survival. It also includes the understanding of ecosystem regulation services value, supply services value, and cultural services value.

### 2.2. Theoretical Analysis

#### 2.2.1. The Direct Influence of Ecological Value Cognition on Pro-Environmental Behavior

Farmers tend to select the environmental behavior that fits their own value judgement standards [40]. According to perceived value theory, individuals will evaluate costs and benefits based on their own cognition before making a decision [41]. As rational economic individuals, farmers will decide their behavior based on utility maximization [42]. If farmers’ ecological value cognition is high, they can foresee that pro-environmental behavior could increase their income, and protecting the environment will increase the possibility of applying pro-environmental behavior [15]. In addition, the Theory of Planned Behavior (TPB) has been one of the most important theories regarding the generation of individual behavior in social psychology [43]. It provided a specific analytical model and paradigm to explain individual behavior and revealed the generation mechanism and reason behind the behavior. Many agricultural economists and social psychologists applied TPB to the pro-environment research field, which has been aptly verified in practice. TPB had been proven to be highly applicable in the use of pesticides [44], farmers’ intentions regarding straw recycling [45], and farmers’ intentions to conserve on-farm biodiversity [46]. According to TPB, individuals form their cognition by screening gathered information and selecting their behavior according to cognition [1]. In addition, farmers are not only rational economic individuals but also rational ecological beings. According to the theory of ecological economic humans in ecological economics, people in the ecological economic system benefit from both the economic system and the ecosystem, as well as the coordination between the two [47,48]. Economic rationality meant that individual economic profit was the only purpose, which transformed the relationship between man and nature into the relationship of that between people and tools [49]. Ecological rationality was the scientific cognitive ability of people regarding the ecological environment, which made up for the lack of economic rationality to the cognition of natural value and reflected the utility level that farmers could obtain from pro-environmental behavior [50,51]. In recent years, work on rural revitalization has strengthened farmers’ awareness of ecological protection and promoted the formation of farmers’ ecological rationality. The government improved farmers’ recognition of rural ecological environment protection through economic means mainly based on subsidies while also shaping farmers’ ecological rationality through training, demonstration, mobilization, information dissemination, and other measures. Therefore, farmers have both economic rationality, which pays attention to agricultural production efficiency, and ecological rationality, which focuses on ecological value. Farmers can pursue the goal of maximizing the long-term benefits of the entire ecological economic system and seek a balanced support point between ecological protection and economic development, as well as between short-term and long-term benefits [51]. Alongside the development of society and the improvement of environmental management, farmers’ cognition regarding ecological value was enhanced. This caused farmers to consider environmental protection while pursuing economic benefits, therefore reducing the risk induced by missed operations and income uncertainty and increasing the possibility of applying pro-environmental behaviors [23,24]. In other words, the higher the ecological value cognition, the higher the possibility of applying pro-environmental behavior. In this case, we pose Hypothesis 1 (H1): Ecological value cognition positively influences pro-environmental behavior (Figure 1).

**Hypothesis** **1** **(H1):**
*Ecological value cognition positively influences pro-environmental behavior.*


**Hypothesis** **2a** **(H2a):**
*Ecological value cognition positively influences environmental attitude.*


**Hypothesis** **2b** **(H2b):**
*Environmental attitude positively influences pro-environmental behavior.*


**Hypothesis** **2** **(H2):**
*Environmental attitude has a mediating effect on the influence of ecological value cognition on pro-environmental behavior.*


**Hypothesis** **3** **(H3):**
*Place identity has a moderating effect on the influence of environmental attitude on pro-environmental behavior.*


**Hypothesis** **4** **(H4):**
*Place dependence has a moderating effect on the influence of environmental attitude on pro-environmental behavior.*


#### 2.2.2. The Mediating Effect of Environmental Attitude

Environmental attitude refers to the general attitude held by individuals when they face environmental problems [52]. Attitude is the expression or secondary consequence of value cognition [40]. The formation of individuals’ environmental attitude is influenced by their cognition of ecological value [53,54]. Therefore, when farmers have high ecological value cognition, they realize that the ecosystem can create value. Farmers’ general views on the rural natural environment and the relationship between people and the environment will change from negative to positive. Positive emotions such as environmental belief and environmental sensitivity will increase. In this case, we pose Hypothesis 2 (H2): Ecological value cognition positively influences environmental attitude.

Environmental attitude greatly influences pro-environmental behavior in a direct way. Many studies have suggested that environmental attitude is an important factor influencing pro-environmental behavior. A series of research frameworks have been proposed based on “Attitude Influence Behavior” theory. Attitude is an important factor that determines behavior in both the Plan–Behavior Theory [55] and the Attitude–Behavior–Cognition Theory [56,57]. The more positive the attitude towards certain behavior, the higher the possibility that individuals will apply that behavior [58,59]. When applying this theory to farmers’ pro-environmental behavior, environmental attitude likely influences pro-environmental behavior [60]. In this case, we pose Hypothesis 2b (H2b): Environmental attitude positively influences pro-environmental behavior.

In social psychology studies, the Value–Attitude–Behavior model (VAB) has been widely used to explain behavior [61]. The VAB model has pointed out that attitude has a mediating effect on the influence of value cognition on behavior [53,62]. In this case, combined with H2a and H2b, we pose hypothesis H2: Farmers’ environmental attitude has a mediating effect on the influence of ecological value cognition on pro-environmental behavior (Figure 1).

#### 2.2.3. The Moderating Effect of Place Attachment in the Influence of Environmental Attitude on Pro-Environmental Behavior

Place attachment refers to the emotional connection between individuals and their residential area [63]. Place is the area where individuals feel their existing value, which is rural areas for farmers [64]. Hometown feelings represent a particular special emotion in China’s rural areas. Traditional customs and cultures such as “Be attached to one’s native land and unwilling to leave it”, and “Fallen leaves return to the roots” have caused Chinese farmers to become attached to their residential area [32,65]. This kind of place attachment of farmers to rural areas in their hearts can be externalized into pro-environmental behavior [66]. Place attachment could evoke farmers’ love for their rural area and encourage them to exhibit positive environmental attitudes. This could reduce the self-interest mentality of farmers when they face environmental problems and promote farmers to reduce short-sighted ideas and behaviors with the cost of sacrificing the environment. More concern for and investment into pro-environmental behavior will be performed [67,68]. In contrast, if farmers show low place attachment, they will continue their former behavior manner, which may damage the environment. In this case, pro-environmental behavior will not be performed even if farmers have a positive attitude [69]. Therefore, place attachment could be a potential factor that influences the process of environmental attitude affecting pro-environmental behavior. Under the same environmental attitude, the farmers that have higher place attachment will perform more obvious pro-environmental behavior than those who have lower place attachment.

Williams’ place attachment theory states that place identity and place dependence are two dimensions of place attachment [70,71]. Place identity refers to the common value that farmers share with each other. This is considered farmers’ emotional support. Place dependence means that farmers’ functional demand needs to be satisfied by the natural environment and the social-cultural environment. Place identity is one type of emotional attachment, while place dependence is one type of functional attachment [70,72]. Different types of place attachment may have different moderating effects on the interaction between environmental attitude and pro-environmental behavior.

(1) The moderating effect of place identity

Social networks that have high place identity are rather stable. Therefore, remaining consistent with the group is an important feature of farmers regarding place identity [72]. According to Social Comparison Theory, when farmers face the situation of pro-environmental behavior selection, most of them will make decisions based on the majority selection and concern that their attitude and behavior are approved by others [67,73]. When farmers hold an active environmental attitude and notice that their community performed pro-environmental behavior, herd mentality will cause them to perform pro-environmental behavior more actively [58]. Furthermore, when farmers hold an active environmental attitude but do not have place identity, their enthusiasm to perform pro-environmental behavior will decrease. Therefore, place identity moderates the relationship between environmental attitude and pro-environmental behavior. Based on the above analysis, we pose Hypothesis 3 (H3): Place identity has a moderating effect on the influence of environmental attitude on pro-environmental behavior. (Figure 1).

(2) The moderating effect of place dependence

According to the Appraisal Theory of Emotion, when individual benefits are satisfied, active attitude and behavior will be evoked [67,74]. Therefore, farmers will judge whether their environmental attitude and pro-environmental behavior are good for themselves or not and then make a decision. The functional element of rural areas is the most intuitive feeling in farmers’ production activities and lives [72]. Farmers that have high place dependence will focus more on the improvement of the functional element. Since an active environmental attitude and pro-environmental behavior could promote the improvement of production and living conditions, the environmental attitude of farmers who have higher dependence will show more influence on pro-environmental behavior. Based on the above analysis, we pose Hypothesis 4 (H4): Place dependence has a moderating effect on the influence of environmental attitude on pro-environmental behavior (Figure 1).

## 3. Study Design 

### 3.1. Study Area

We selected Jinan as the research area. Jinan is the capital city of Shandong Province in China. The average GDP per person is 106,416.00 Yuan (RMB), which is higher than the national average level (71,999.60 Yuan). A total population of 2,452,800 lives in 5530 villages in Jinan. The village population accounts for 36.95% of the total city population, which is higher than the average national village population percentage (36.11%). Jinan has good economic development conditions and a large rural population. In recent years, Jinan has invested 13.267 billion Yuan in rural areas to promote rural environmental improvement. In Jinan, 120 beautiful model villages have been built, and 3135 clean villages have been created, accounting for 66.7%. At the same time, Jinan has been ranked first in the assessment of rural human settlements in Shandong Province for two consecutive years. Therefore, Jinan is a good study area to discuss how to enhance farmers’ pro-environmental behavior.

### 3.2. Participants

The data were collected from a questionnaire. “Farmer’s pro-environmental behavior questionnaire in Jinan” was designed with 56 related questions. Stratified sampling and random sampling were used for the study. First, the district was set as the elementary sampling unit. Lixia, Shizhong, Licheng, and Changqing districts were selected as the study area according to the economic conditions, of which Lixia district is the most developed area, Changqing is the most undeveloped district, and the other districts exist between them. Secondly, 40 to 60 farmers were selected for one-on-one interviews in 8 randomly selected villages in each district to conduct the survey. The survey was divided into two parts: The pre-survey (June to July 2021) and the formal survey (August to October 2021). In the pre-survey, 100 questionnaires were distributed to improve the questionnaire. In the formal survey, 850 questionnaires were distributed, 681 of which were valid. Descriptive statistic results are shown in Table 1. 

### 3.3. Instruments

The scale tables of ecological value cognition, environmental attitude, place attachment, and pro-environmental behavior were designed based on the existing maturity scale table and our research purpose. A 5-point Likert scale was used to assign values to each variable, in which 1, 2, 3, 4, and 5 corresponded to totally disagree, disagree, do not care, agree, and totally agree, respectively.

#### 3.3.1. Independent Variable: Ecological Value Cognition

According to the “System of Environmental-Economic Accounting 2012—Central Framework” [38], “Ecosystems, Human Well-being: A Framework for Assessment” [75], and “Living Beyond Our Means: Natural Assets and Human Well-Being--Statement from The Board” [76], the ecological value cognition scale table was created from the dimensions of regulating services value, provisioning services value, and cultural services value (Table 2).

#### 3.3.2. Dependent Variable: Pro-Environmental Behavior

The scale table of pro-environmental behavior was designed from the angle of ecological management, consumption behavior, persuasive behavior, and citizenship behavior according to Sia A.P., et al. (1986) [77], Smith-Sebasto, N.J., et al. (1995) [78], and Kaiser F.G., et al. (2003) [79,80] (Table 3). Farmers’ living conditions and habitual differences between China and western countries were also considered. Ecological management refers to the practical activities that maintain and improve the current ecosystem. Consumption behavior is the behavior that uses economic means to protect the environment. Persuasive behavior is the behavior that persuades people to protect the environment, which can change the beliefs and values of humans. Citizenship behavior represents protecting the environment by supporting or accepting public policy (Table 3).

#### 3.3.3. Mediator Variable: Environmental Attitude

The scale table of environmental attitude was designed from the dimensions of environmental belief and environmental sensitivity based on the NEP scale table developed by Dunlap [81]. The studies of Hsu S.J. et al. (1998) [82], Stern P. C. et al. (1999) [83], and Chan R.Y.K. et al. (2001) [84] were also taken as references. Environmental belief refers to the common opinion that individuals hold towards the natural environment and the relationship between humans and the environment. Environmental sensitivity refers to emotions such as enjoyment, concern, and sympathy that individuals have toward the environment (Table 4).

#### 3.3.4. Moderator Variable: Place Attachment

The scale table of place attachment was designed according to Williams’s two-dimensional scale [71,85] (Table 5). Meanwhile, the studies of Song et al. (2018) [86], Sebasto, N.J.S et al. (2010) [78], Chen et al. (2018) [85], and Plunkett et al. (2019) [87] were also considered. The real situation of farmers and the rural area conditions in China were taken into consideration as well.

### 3.4. Procedure

All study procedures were approved by the researchers’ institutional review board and the school’s administration. On the day of farmers’ data collection, trained study staff (graduate researchers and undergraduate research assistants) conducted random interviews with villagers. All 850 (100%) farmers provided consent and completed the survey. Survey questions were read loudly by researchers while farmers responded on their own paper copy. Other members of the research team were available to answer questions and provide assistance when needed. Surveys were generally completed within 30 min. For farmer participation, each farmer received a $1.5 donation for daily use.

### 3.5. Data Analysis

SPSS 21.0 (IBM, New York, NY, USA) and AMOS 24.0 (IBM, New York, NY, USA) were used for data analysis in this study. First, the reliability and validity of the scale table were examined. Second, a structural equation model was applied to examine the pathway through which ecological value cognition and environmental attitude act on pro-environmental behavior. The mediating effect of environmental attitudes was also examined. Finally, the moderating effects of place identity and place dependence on the influence of environmental attitude on pro-environmental behavior were examined.

## 4. Results

### 4.1. Structural Equation Model Test

#### 4.1.1. Reliability and Validity Analysis

##### Reliability Test

After a reliability analysis of 19 items for the independent variable of ecological value cognition, 16 items for the dependent variable of pro-environmental behavior, 7 items for the mediator variable of environmental attitude, and 9 items for the moderator variable of place attachment, the Cronbach α coefficients were found to all be greater than 0.7, indicating that this part of the questionnaire had good reliability (Table 6).

##### Validity Test

We used confirmatory factor analysis to test the validity. To determine whether the confirmatory factor analysis model was valid, we considered various fitting indicators. The results show that the χ^2^/df values of all variables were less than 3. The GFI, AGFI, NFI, TLI, and CFI of all variables were greater than 0.9, indicating that the model had good adaptability. The RMSEA of all variables was less than 0.08, indicating that the adaptability was good, and the model fitting degree met the requirements (Table 7).

As can be seen from Table 8, the standardized factor loading of each question item ranged from 0.663 to 0.924, which were all greater than 0.5, and the standard error was all less than the standard value of 0.5, indicating that each question item could explain its dimension well, which echoed the results of exploratory factor analysis and further proved that the validity of the questionnaire was good. The combined reliability CR values were all greater than 0.7, indicating that all the items in each latent variable could consistently explain the latent variable. The AVE values were all above the standard value of 0.5, indicating that the scale table used in this paper had good convergent validity.

#### 4.1.2. Structural Equation Model

The result showed that the χ^2^/df value (2.144) is smaller than 3 and the RMSEA value (0.041) is smaller than 0.08. This indicated the good fit of the model. GFI (0.895), AGFI (0.882), NFI (0.900), TLI (0.940), and CFI (0.944) fit the generic standard, which indicates the structural equation model constructed in this study is valid and fit our data well (Table 9).

### 4.2. Pathway Analysis

The result of the pathway analysis showed that the normalized path coefficient of ecological value cognition on pro-environmental behavior is 0.726 (C.R. = 5.431, *p* = 0.000 < 0.001). It indicates that ecological value cognition has a significant positive influence on pro-environmental behavior, which supports hypothesis H1. The normalized path coefficient of ecological value cognition on environmental attitude is 0.876 (C.R. = 13.322, *p* = 0.000 < 0.001). This indicates that ecological value cognition has a significant positive influence on environmental attitude, which supports hypothesis H2a. Similarly, the normalized path coefficient of environmental attitude on pro-environmental behavior is 0.373 (C.R. = 2.965, *p* = 0.003 < 0.001). This indicates that environmental attitude has a significant positive influence on pro-environmental behavior, which supports hypothesis H2b (Table 10, Figure 2).

### 4.3. Mediating Effect Test of Environmental Attitude

Bootstrap in AMOS was used to conduct a mediating effect test. With 2000 repeats (95% confidence), the result showed that the value of the effect of the mediating path (ecological value cognition–environmental attitude–pro-environmental behavior) is 0.327, which is between 0.042 and 0.600 (the upper and lower intervals of 95% confidence) (*p* < 0.05). This indicated the existence of a mediating effect, which supports hypothesis H2 (Table 11).

### 4.4. Moderating Effect Test of Place Attachment

Three multiple regression equations were built using multiple hierarchical regression analysis to test the regulating effect. The control variable was introduced in the first model. The independent variable, regulating variable, and interaction term were controlled to avoid pseudo regression. The control variable, independent variable, and regulating variable are all introduced in the second model. This model is built to determine whether the independent variable and regulating variable will influence the dependent variable through the explanation ability (R^2^ value). The control variable, independent variable, moderating variable, and interaction term were introduced in the third model. The significant regression coefficient and increased R^2^ of the interaction term could indicate a regulating effect. 

#### 4.4.1. Moderating Effect of Place Identity 

In Table 12, gender, age, income per month, and education were set as control variables. In model 1, the control variable was set as the independent variable and pro-environmental behavior was set as the dependent variable to build the multiple regression model. Under the control of the control variable, model 2 set environmental attitude and place identity as independent variables and pro-environmental behavior as the dependent variable to build the model. In model 3, under the control of the control variable, environmental attitude, place identity, and environmental attitude*place identity (interaction term) were set as the independent variables and pro-environmental behavior was set as the dependent variable.

The result showed that environmental attitude had a significant positive influence on pro-environmental behavior in model 2 (β = 0.67, *p* < 0.001). The regression coefficient of the interaction term in model 3 is 0.098 (*p* < 0.001), indicating a significant positive influence of the interaction term on pro-environmental behavior. R^2^ in model 3 (0.491) is larger than that in model 2 (0.482). This result indicates increased explanation ability (Table 12).

In order to show the real trend of the moderating effect clearly, a simple slope test was conducted after the multiple regression analysis (Figure 3). The result showed that environmental attitude had a positive influence on pro-environmental behavior. As environmental attitudes increased, pro-environmental behavior also increased. When place identity was low, the slope was relatively gentle (Figure 3, full line). When place identity was higher, the slope increased (Figure 3, dashed line). The above result indicates that under strong place identity moderation, the positive effect of environmental attitude on pro-environmental behavior was strong. 

The above result indicates that the influence of environmental attitude on pro-environmental behavior will increase along with farmers’ increased place identity. In other words, place identity has a positive moderating effect on the influence of environmental attitude on pro-environmental behavior, which supports hypothesis H3.

#### 4.4.2. Moderating Effect of Place Dependence 

In Table 13, gender, age, income per month, and education were set as control variables. In model 1, the control variable was set as the independent variable and pro-environmental behavior was set as the dependent variable to build a multiple regression model. Under the control of the control variable, model 2 set the environmental attitude and place dependence as the independent variables and pro-environmental behavior as the dependent variable to build the model. In model 3, under the control of the control variable, environmental attitude, place dependence, and environmental attitude*place dependence (interaction term) were set as independent variables and pro-environmental behavior was set as the dependent variable.

The result showed that environmental attitude has a significant positive influence on pro-environmental behavior in model 2 (β = 0.693, *p* < 0.001). The regression coefficient of the interaction term in model 3 is 0.096 (*p* < 0.001), indicating the significant positive influence of the interaction term on pro-environmental behavior. The R^2^ value in model 3 (0.486) is larger than that in model 2 (0.478). This indicates increased explanation ability (Table 13).

Similarly, in Figure 4, under strong moderation of place dependence, the positive effect of environmental attitude on pro-environmental behavior was also strong. The result showed that environmental attitude had a positive influence on pro-environmental behavior. As environmental attitudes increased, pro-environmental behavior also increased. When place dependence was low, the slope was relatively gentle (Figure 4, full line). When place dependence was higher, the slope increased (Figure 4, dashed line). The above result indicates that under strong place dependence moderation, the positive effect of environmental attitude on pro-environmental behavior was strong (Figure 4).

The above result indicates that the influence of environmental attitude on pro-environmental behavior will increase alongside farmers’ increased place dependence. In other words, place dependence has a positive moderating effect on the influence of environmental attitude on pro-environmental behavior, which supports hypothesis H4.

## 5. Discussion

### 5.1. The Direct Influence of Ecological Value Cognition on Pro-Environmental Behavior 

Regarding hypothesis H1, this study proved ecological value cognition has a direct positive effect on pro-environmental behavior. This result is consistent with Kotchen, M.J. (2000) [16], Halkos, G. (2014) [17], and Duan, P. (2021) [18]. The significant positive effect of ecological value cognition on pro-environmental behavior is likely due to the following reasons. 

First, farmers realized the multifaceted value of the ecosystem, which promoted the occurrence of pro-environmental behavior. Ecosystems have provisioning services value, regulating services value, and cultural services value, of which provisioning services are related to the material benefit of environmental property, showing the attribute of personal items, which could be traded in the market. This result reflects the concept of exchange value. Regulating services and cultural services are related to the non-material benefits of environmental property, showing the attribute of public goods. This result reflects the concept of welfare economic value. Furthermore, ecosystem service value has three obvious features: Useful to humans (use value), currently in short supply or will be in the near future, and is able to enter human economic activity. When ecological resources enter human activity, part of them are labor-joined resources, including their use value and attached value due to labor participation. These kinds of ecosystem resources have been shown as products, functions, benefits, etc. The remainder represents ecological resources that, without labor participation, have virtual value, shown as use value and exchange value. Therefore, farmers who are economically rational and ecologically rational can easily realize the market value and non-market value of the ecosystem if they have high ecological value cognition. Farmers tend to perform pro-environmental behavior if they realize that pro-environmental behavior could produce more benefits. Second, Jinan is a developed city in China, and therefore farmers have a strong sense of ecological protection. According to Maslow’s hierarchy of needs theory, the better the economy, the higher the emotional demand. As Jinan is the capital city of eastern China, the economy in rural areas is relatively better than in other cities. Farmers usually do not move out. Therefore, farmers’ demand on the environment is high. Meanwhile, farmers also have a strong sense of environmental protection. The higher ecological value cognition that farmers have promoted the occurrence of pro-environmental behavior.

### 5.2. The Mediating Effect of Environmental Attitude

Regarding hypothesis H2, our results indicate that in the process of ecological value cognition influencing pro-environmental behavior, environmental attitudes play a mediating role. As farmers’ ecological value cognition increases, their environmental attitude will be more active, thus the possibility of performing pro-environmental behavior will increase. This finding proved that the VAB model is reasonable [61]. The improvement in farmers’ environmental attitude may be due to the rural environmental regulation of Jinan in recent years. The government of Jinan has taken on the improvement of the rural living environment as a key task in the implementation of its rural revitalization strategy. In recent years, 13.267 billion yuan was invested in rural areas to promote the improvement of the rural living environment. At the same time, 120 beautiful rural model villages and 3135 clean villages were built. The government of Jinan follows the principle of “large investment and high standards” and insists on the integrated promotion of residential environment improvement with industrial development, rural management, and civilization promotion.

The government of Jinan has made great efforts to ameliorate rural human settlements, which has improved farmers’ environmental attitudes. First, the appearance level of the countryside has improved through a variety of ways to create beautiful countryside. During the three-year plan for rural living environment improvement, 9.931 billion in special funds has been invested and another 3.336 billion yuan has been invested in rural construction since 2021. In order to build model villages, 111 standards were issued. Second, The management and protection mechanism has been continuously improved. For example, in order to promote the classified management of rural waste, the “Jinan Municipal Domestic Waste Reduction and Classification Management Regulations” was promulgated. In terms of the toilet revolution, the existing long-term mechanisms of toilet maintenance, feces and sewage removal, and resource utilization have been improved. In the sewage treatment aspect, according to the requirements of “integrated construction and management, professional operation and maintenance”, it is proposed that the city and district (county) finance will compensate the rural operation and maintenance fees. Third, a variety of policies have been proposed to encourage farmers to actively improve the rural environment. The government provides free cement, sand, and other raw materials and greening seedlings to encourage farmers to build roads and plant trees to beautify the countryside. In addition, through grid management, integral system management, and other measures, the enthusiasm and sense of responsibility of farmers have been evoked to participate in the renovation of the human living environment, 

The mediating effect of environmental attitude likely stems from the fact that when farmers have emotions of active concern, enjoyment, and sympathy for the environment, they will hold active attitudes toward environmental problems as well. In other words, farmers may have high environmental beliefs and sensitivity. Therefore, farmers’ ecological value cognition could pose a stronger influence on pro-environmental behavior through the mediation of environmental attitudes.

### 5.3. The Moderating Effect of Place Attachment in the Influence of Ecological Value Cognition on Pro-Environmental Behavior

In contrast to the research results of Wang, X., et al. (2021) [66], Wan, C., et al. (2021) [67], and Budruk, M., et al. (2009) [68], we found a moderating effect of place identity and place dependence on the influence of ecological value cognition on pro-environmental behavior. The possible reasons may be the following. 

Regarding hypothesis H3, place identity may have a moderating effect because, as one type of prosocial behavior, pro-environmental behavior has the attributes of altruism. This kind of behavior is good for other people and could enhance public welfare. However, farmers cannot directly benefit from this behavior, which requires the collective action of villagers to produce substantial results. Another reason is that farmers generally have a herd mentality. Regardless of farmers’ views on improving environmental behavior, they may take action consistent with those of most farmers [69]. The validation of hypothesis H3 means farmers’ environmental attitude and pro-environmental behavior could be accepted by their community, which causes the influence of environmental attitude on pro-environmental behavior to become stronger.

Regarding hypothesis H4, the moderating effect of place dependence means the production and living environment that villages provide could satisfy farmers’ demands. Stemming from egoism, farmers prefer a better ecological environment in their villages. This kind of demand could enhance the moderating effect of environmental attitudes on pro-environmental behavior.

In Jinan, rural revitalization has proceeded actively in recent years. First, in terms of social assistance, more than 180,000 people living in poverty were included in the social assistance system. At the same time, 1.284 billion yuan were arranged to support 381 assistance projects. Second, great efforts were made to improve the income level of farmers. The disposable income of rural residents per capita was 22,580 yuan, which causes the income ratio of urban and rural residents to drop to 2.54. The proportion of villages with a collective income of more than 100,000 yuan rose to 91.3%. Third, basic public services in rural areas have been strengthened. In terms of education, rural primary and secondary schools and kindergartens were constructed according to the unified standards for urban and rural areas. In total, 1259 rural welfare houses were built, and national basic public health service projects covered 52 town hospitals and 3679 village clinics. The basic pension standard, government subsidy standard for basic medical insurance, financing standard for serious illness insurance, and social assistance standard for farmers increased simultaneously. In this process, farmers’ place identity and place dependence have been strengthened, thus moderating the impact of farmers’ environmental attitudes on pro-environmental behavior.

## 6. Conclusions

Jinan was selected as the case study area in this study. The influence mechanisms of ecological value cognition, environmental attitude, and place attachment (place identity and place dependence) on pro-environmental behavior were discussed. The result showed that there are two pathways in the influence of ecological value cognition on pro-environmental behavior, the first in which ecological value cognition directly influences pro-environmental behavior, and the second in which ecological value cognition indirectly influences pro-environmental behavior through the mediating effect of environmental attitude. Meanwhile, place identity and place dependence showed a moderating effect on the influence of environmental attitude on pro-environmental behavior. The following suggestions are proposed based on the above conclusions. 

Local governments could increase farmers’ ecological value cognition through education. First, governments could advertise knowledge of ecological value, environment protection, and sustainable development of the environment to improve farmers’ ecological value cognition and inspire them to actively perform pro-environmental behavior. Second, the government can correct farmers’ cognitive bias through relevant technical training and advertising long-term economic benefits and environmental benefits brought by green production technologies to improve farmers’ enthusiasm and initiative in adopting environmentally friendly technologies. 

Local government could also cultivate farmers’ active environmental attitudes through advertising and education. First, stories, cartoons, and pictures could be used to stir farmers’ environmental beliefs and sensitivity. Meanwhile, typical case displays, good examples of learning, visiting, and live demonstrations are all good strategies to evoke farmers’ active emotions toward pro-environmental behavior, which also enhances their sense of awe regarding the ecological environment. Second, governments could focus on enhancing farmers’ subjective constraints, inspiring them to perceive themselves as the main person responsible for environmental protection. Governments could emphasize individual farmers’ obligations to enhance their environmental beliefs and sensitivity.

Local governments should enhance farmers’ place attachment, strengthen guided environmental education, and encourage farmers to supervise each other. First, governments could advertise the representative accomplishment that a village has achieved and its historical cultural resources and hold cultural activities to enhance farmers’ place identity. Second, governments should invest more funds to improve infrastructure construction to increase farmers’ place identity and place dependence. Third, the government should focus on strengthening the guided education of pro-environmental behavior. The government can strengthen the local recognition of pro-environmental behavior by setting up demonstration villages and households.

Local governments could formulate reward and punishment measures to encourage and push farmers to perform pro-environmental behavior. First, governments could establish a subsidy system for pro-environmental behavior and introduce more social capital to promote the spread of pro-environment technology to reduce the risk and cost when farmers perform pro-environmental behavior. Second, governments should extend the supervision scope of non-environmentally friendly industries and increase punishment in order to further guide farmers’ pro-environmental behavior. 

## Figures and Tables

**Figure 1 ijerph-19-17011-f001:**
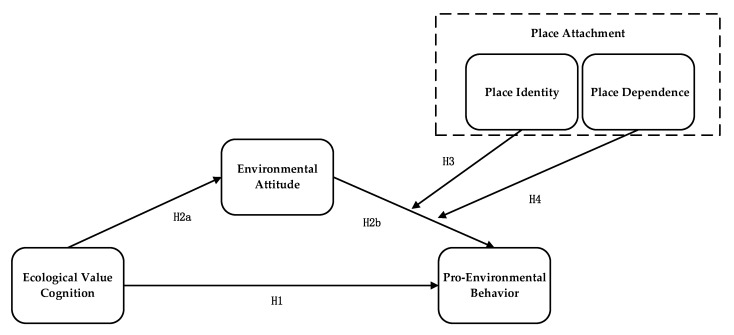
Theoretical analysis model.

**Figure 2 ijerph-19-17011-f002:**
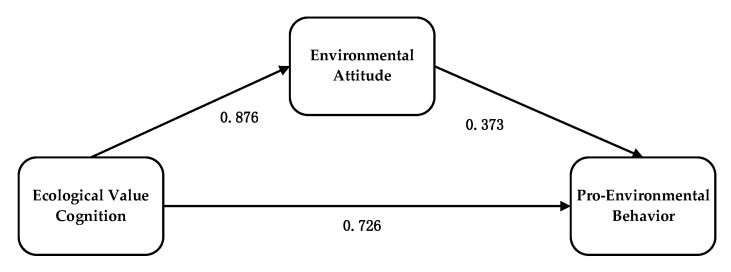
Pathway analysis result.

**Figure 3 ijerph-19-17011-f003:**
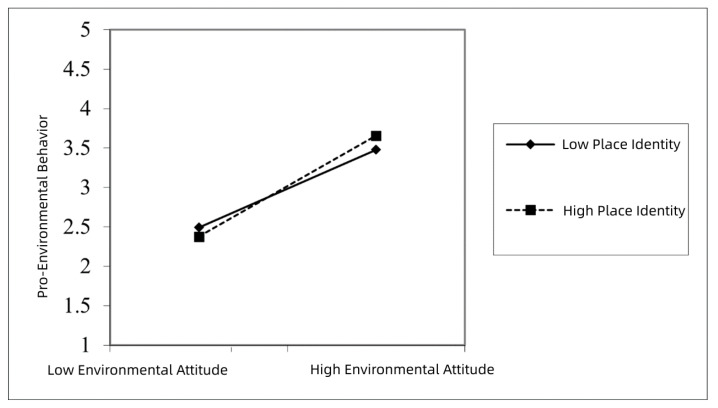
The moderating effect of place identity.

**Figure 4 ijerph-19-17011-f004:**
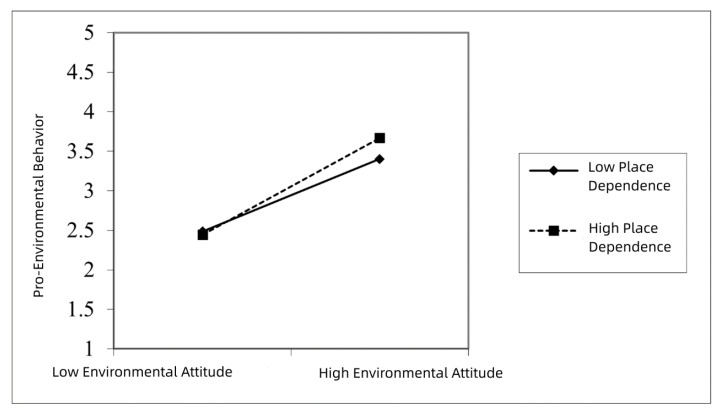
Moderating effect of place dependence.

**Table 1 ijerph-19-17011-t001:** Descriptive statistic results.

Basic Information	Criteria	Frequency	Percentage
Gender	Male	390	57.3
	Female	291	42.7
Age	≤18	55	8.1
	19–30	255	37.4
	31–40	196	28.8
	41–50	145	21.3
	51–60	24	3.5
	≥61	6	0.9
Income per month	<4000	78	11.5
	4000–5500	176	25.8
	5000–6000	229	33.6
	6000–7500	124	18.2
	≥7000	74	10.9
Highest education	Junior middle school and below	97	14.2
	High school, special school, and technical school	121	17.8
	Junior college	160	23.5
	Undergraduate	227	33.3
	Junior middle school and below	97	14.2

**Table 2 ijerph-19-17011-t002:** Scale table of ecological value cognition.

Dimension	Index
Regulating service value cognition	A1. I think ecological environment could regulate air quality (for example, ecosystems can not only release chemical substances into the atmosphere, but also absorb chemical substances from the atmosphere, thus affecting air quality).
	A2. I think ecological environment could regulate climate (for example, in small scale, the change of landscape could influence air temperature and rain fall; in global scale, ecosystem play important role through store and release greenhouse gases).
	A3. I think ecological environment could regulate water resource (for example, the change of landscape could influence surface runoff, flood, and water retain in the aqueous stratum).
	A4. I think ecological environment could regulate erosion (for example, Vegetation cover could help to maintain soil and avoid collapse).
	A5. I think ecological environment can purify water and handle wastes (for example, ecosystem could help to filter and decompose organic waste that enter freshwater and marine ecosystem).
	A6. I think ecological environment can regulate disease (for example, the change of ecosystem could indirectly change the amount of human pathogens and vectors).
	A7. I think ecological environment can regulate insect pest (for example, the change of ecosystem could influence the diffusion of farmland pest).
	A8. I think ecological environment can help pollination (for example, ecosystem change could influence the distribution and abundance of pollination medium).
Provisioning value cognition	A9. I think ecological environment can provide food for human being (for example, plant, animal and micro-organisms could provide food for human).
	A10. I think ecological environment can provide fiber for human being (for example, wood, jute, cotton, silk and wool).
	A11. I think ecological environment can provide fuel (for example, wood that could be used as resource, raw material like livestock excrement).
	A12. I think ecological environment can provide bio-chemical material, natural medical materials, and medicine (for example, bio-pesticide and food additive).
	A13. I think ecological environment can provide decorative resources (for example, skin, shell, and flower could be used as decoration while whole plant could be used to beautify landscape).
	A14. I think ecological environment can provide freshwater to human being.
Culture value cognition	A15. I think ecosystem diversity can influence culture diversity.
	A16. I think the spirit of man many religion originated from ecosystem.
	A17. I think the component of ecosystem and formation process could provide material for education.
	A18. I think ecosystem provide inspiration source for art, folk legend, national symbol, architecture style and advertisement design.
	A19. I think human will select the place to visit partly based on the ecological environment feature.

**Table 3 ijerph-19-17011-t003:** Pro-environmental behavior scale table.

Dimension	Index
Ecological management	B1. I save resource (for example, close the light when stop to use it; walk instead of driving; use air-condition properly).
	B2. I save water (for example, use instrument that can save water; close the tap water when do not use it; reuse water for irrigation, and reduce the time when taking shower).
	B3. I don’t use product that produce for only one time using (plastic bag, one time use tableware, and paper cup).
	B4. I don’t discard garbage carelessly in public place even there have no dustbin nearby.
	B5. I try to improve surrounding environment start from myself (for example, plant tree, clear the road and playground).
Consumption behavior	B6. I don’t buy product that are not environment friendly (for example, life product which is not environment friendly, fertilizer that have heavy pollution).
	B7. I purchase energy saving household appliances (for example, energy saving refrigerate, air-condition and wash machine).
	B8. I purchase product that with symbol of green product even they may be more expensive.
Persuade behavior	B9. I encourage other people to protect environment (for example, saving resource and water, garbage recycle).
	B10. I persuade other people to stop the behavior that damage environment (for example, discard garbage carelessly, and release polluted waste water).
	B11. I tell the importance of environment protection to other people.
	B12. I express my support to environment protection publicly (give speech, accept interview, and publish articles).
Citizenship behavior	B13. I participate environment protection related meeting.
	B14. I report people or units that damage the environment to relevant units.
	B15. I advertise to other people about environment protection related law.
	B16. I discuss with other people about how to solve environment problem.

**Table 4 ijerph-19-17011-t004:** Environmental attitude scale table.

Dimension	Index
Environmental belief	C1. I think human does not have the right to change natural to satisfy our own demand.
	C2. I think the existing of animals and plants is not only for human use, but also for maintaining ecological balance.
	C3. I think the self-balance ability of nature could not deal with the influence induced by the development of modern industry, we must protect environment.
	C4. I love nature, and i am interested in nature.
Environmental sensitivity	C5. I have the obligation to protect environment, and i can scarify my own benefit and change my life style for environment protection.
	C6. I have the obligation to protect environment, and i can offend some people if necessary.
	C7. I am angry with the environment problem that the media reported.

**Table 5 ijerph-19-17011-t005:** Place attachment scale table.

Dimension	Index
Place Identity	D1. I think i am one of the community members in our village.
	D2. I have a strong sense of identity with the village.
	D3. I have a strong sense of belonging to the village where i can understand myself better.
	D4. I agree with and accept the traditional customs of the village.
	D5. I have common value sense with most of our community member in our village.
Place dependence	D6. Currently, my village is best working and living place for me.
	D7. My village provide life condition that other place could not provide.
	D8. My village have great life guarantee system.
	D9. If i got chance to select freely, i would not give up living in my village.

**Table 6 ijerph-19-17011-t006:** Reliability analysis of variables.

Variables	Dimension	Number of Items	Cronbach’s α
Ecological Value Cognition	Regulating service value cognition	8	0.9
	Provisioning value cognition	6	0.878
	Culture value cognition	5	0.882
Pro-Environmental Behavior	Ecological management	5	0.845
	Consumption behavior	3	0.852
	Persuade behavior	4	0.884
	Citizenship behavior	4	0.844
Environmental Attitude	Environmental belief	4	0.866
	Environmental sensitivity	3	0.86
Place Attachment	Place Identity	5	0.848
	Place Dependence	4	0.824

**Table 7 ijerph-19-17011-t007:** Confirmatory factor analysis model fitting indicators.

Variables	χ^2^/df	GFI	AGFI	NFI	TLI	CFI	RMSEA
Ecological Value Cognition	3.416	0.931	0.912	0.929	0.941	0.948	0.06
Pro-Environmental Behavior	2.425	0.958	0.942	0.955	0.967	0.973	0.046
Environmental Attitude	2.441	0.987	0.973	0.989	0.989	0.993	0.046
Place Attachment	2.259	0.981	0.968	0.975	0.98	0.986	0.043

GFI: Goodness of Fit Index; AGFI: Adjusted Goodness of Fit Index; NFI: Normed Fit Index; TLI: Tucker-Lewis Index; CFI: Comparative Fit Index; RMSEA: Root Mean Square Error of Approximation.

**Table 8 ijerph-19-17011-t008:** Factor analysis results.

Variables	Measurement Topics	Estimate	SE	CR	*p*	CR	AVE
Regulating service value cognition	A1	0.819				0.899	0.529
	A2	0.686	0.044	19.252	***		
	A3	0.686	0.043	19.263	***		
	A4	0.705	0.043	19.916	***		
	A5	0.739	0.044	21.171	***		
	A6	0.706	0.042	19.952	***		
	A7	0.663	0.044	18.453	***		
	A8	0.799	0.042	23.524	***		
Provisioning value cognition	A9	0.757				0.879	0.548
	A10	0.706	0.048	18.262	***		
	A11	0.715	0.048	18.496	***		
	A12	0.74	0.049	19.196	***		
	A13	0.727	0.047	18.852	***		
	A14	0.793	0.046	20.711	***		
Culture value cognition	A15	0.796				0.883	0.602
	A16	0.718	0.044	19.639	***		
	A17	0.774	0.044	21.525	***		
	A18	0.782	0.045	21.794	***		
	A19	0.805	0.043	22.588	***		
Ecological management	B1	0.788				0.847	0.526
	B2	0.679	0.052	17.316	***		
	B3	0.699	0.052	17.852	***		
	B4	0.733	0.054	18.79	***		
	B5	0.723	0.052	18.511	***		
Consumption behavior	B6	0.806				0.852	0.658
	B7	0.827	0.05	21.323	***		
	B8	0.8	0.048	20.879	***		
Persuade behavior	B9	0.855				0.886	0.66
	B10	0.769	0.041	23.134	***		
	B11	0.74	0.04	21.92	***		
	B12	0.878	0.038	27.652	***		
Citizenship behavior	B13	0.802				0.844	0.576
	B14	0.769	0.049	19.942	***		
	B15	0.695	0.046	17.922	***		
	B16	0.766	0.047	19.864	***		
Environmental belief	C1	0.872				0.888	0.667
	C2	0.698	0.036	21.091	***		
	C3	0.752	0.035	23.593	***		
	C4	0.924	0.034	32.132	***		
Environmental sensitivity	C5	0.85				0.861	0.676
	C6	0.875	0.043	24.827	***		
	C7	0.734	0.04	20.894	***		
Place Identity	D1	0.778				0.85	0.532
	D2	0.681	0.05	17.194	***		
	D3	0.672	0.048	16.964	***		
	D4	0.730	0.05	18.502	***		
	D5	0.778	0.048	19.721	***		
Place Dependence	D6	0.773				0.826	0.544
	D7	0.717	0.053	17.346	***		
	D8	0.668	0.052	16.206	***		
	D9	0.785	0.052	18.619	***		

Note: ***, *p* < 0.001, CR, critical ratio (i.e., the t-value corresponding to each item loading); SE, standard error; AVE, Average Variance Extracted.

**Table 9 ijerph-19-17011-t009:** Parameters in structural equation model.

Fitting Index	χ^2^/df	GFI	AGFI	NFI	TLI	CFI	RMSEA
Criteria	<3	>0.8	>0.8	>0.9	>0.9	>0.9	<0.08
Value	2.144	0.895	0.882	0.900	0.940	0.944	0.041

GFI: Goodness of Fit Index; AGFI: Adjusted Goodness of Fit Index; NFI: Normed Fit Index; TLI: Tucker-Lewis Index; CFI: Comparative Fit Index; RMSEA: Root Mean Square Error of Approximation.

**Table 10 ijerph-19-17011-t010:** Pathway analysis result.

Hypothesized Path			Estimate	S.E.	C.R.	*p*
Pro-Environmental Behavior	<---	ecological value cognition	0.726	0.093	5.431	***
Environmental Attitude	<---	ecological value cognition	0.876	0.083	13.322	***
Pro-Environmental Behavior	<---	environmental attitude	0.373	0.069	2.965	0.003 **

Note: ** means *p* < 0.01, *** means *p* < 0.001.

**Table 11 ijerph-19-17011-t011:** Bootstrap for mediating effect test.

Mediating Hypothesis Path	Estimate	S.E.	95% Confidence		*p*
			Lower	Upper	
Ecological Value Cognition-Environmental Attitude-Pro-Environmental Behavior	0.327	0.147	0.042	0.600	0.034 *

Note: * means *p* < 0.05.

**Table 12 ijerph-19-17011-t012:** Moderating effect test of place identity influence on the interaction between environmental attitude and pro-environmental behavior.

Variable	Model 1		Model 2		Model 3	
	β	*t*	β	*t*	β	*t*
Gender	−0.011	−0.296	−0.006	−0.216	−0.008	−0.305
Age	0.041	1.074	0.035	1.257	0.038	1.361
Income Per Month	−0.034	−0.890	0.017	0.591	0.016	0.590
Highest Education	0.008	0.210	−0.028	−0.997	−0.026	−0.946
Environmental Attitude			0.67 ***	22.780	0.665 ***	22.766
Place Identity			0.067 *	2.280	0.081 **	2.758
Environmental Attitude*Place Identity					0.098 ***	3.524
R^2^	0.003		0.482		0.491	
Adjusted R^2^	−0.004		0.476		0.485	
F	0.463		89.392 ***		81.097 ***	

Note: * means *p* < 0.05, ** means *p* < 0.01, *** means *p* < 0.001.

**Table 13 ijerph-19-17011-t013:** Moderating effect test of place identity and place dependence influence on the interaction between environmental attitude and pro-environmental behavior.

Variable	Model 1		Model 2		Model 3	
	β	*t*	β	*t*	β	*t*
Gender	−0.011	−0.296	−0.003	−0.097	−0.002	−0.056
Age	0.041	1.074	0.035	1.253	0.028	1.008
Income per month	−0.034	−0.890	0.019	0.667	0.023	0.812
Highest education	0.008	0.210	−0.028	−0.988	−0.021	−0.746
Environmental Attitude			0.693 ***	23.681	0.703 ***	24.065
Place Identity			−0.005	−0.164	0.02	0.678
Environmental Attitude*Place Identity And Place Dependence					0.096 **	3.292
R^2^	0.003		0.478		0.486	
Adjusted R^2^	−0.004		0.472		0.480	
F	0.463		87.976 ***		79.459 ***	

Note: * means *p* < 0.05, ** means *p* < 0.01, *** means *p* < 0.001.

## Data Availability

The data underlying the results presented in the study are all available. The data presented in this study are available upon request from the corresponding author. The data are not publicly available due to privacy.

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
