# Peer review of "Pro-Environmental Behavior: Examining the Role of Ecological Value Cognition, Environmental Attitude, and Place Attachment among Rural Farmers in China"

_ijerph, 2022, doi:10.3390/ijerph192417011_

Round 1
Reviewer 1 Report (New Reviewer)
Study on the factors that influenced farmer’s pro-environmental behavior is very meaningful. The findings in this study are important to the improvement of rural eco-environment and life quality of farmers. I think it is suitable for publication in International Journal of Environmental Research and Public Health(IJERPH)
There is only one question: please explain in detail why Jinan was chosen as the case area.
Author Response
Please see the attachment.

Reviewer 2 Report (New Reviewer)
I want to thank the opportunity to review the manuscript titled "Pro-Environmental Behavior Examining the Role of Ecological Value Cognition, Environmental Attitude, and Place Attachment among Rural Farmers in China. This article is well-written and may make significant contributions to the field. The framework and methodology were all well-developed. I have a few mirror comments that hopefully, the authors can address in the revised version.
1. Introduction/ Research Gap Justification
The authors mentioned that “the definition is vague which is always mixed with environmental consciousness and environmental concern…(thus), it is necessary to propose a clear definition of ecological value cognition”. More justification is needed to support this statement. Why is it needed? If the conceptualization work is done currently, it could be a significant theoretical contribution made by this study.
The authors also stated that “demographic factors had relatively limited impact on farmers' pro-environmental behavior”. This sentence is not clear. I am not sure what this sentence means and how it is related to research gap justification.
The authors emphasized the need for conducting this research to study farmers’ perceptions and behaviors. Please provide a justification to explain how farmers are different from other subjects.
2. Literature Review
In section 2.1, the authors discuss the definition of ecological value cognition. What is the new definition of ecological value cognition this study has proposed? Based on the previous discussion, readers expect to see a new definition of ecological value cognition. Please expand and highlight.
3. Format and English
I recommend the author summarize the results of SEM using a figure of a structural model.
Professional English editing is needed.
Author Response
Please see the attachment.

This manuscript is a resubmission of an earlier submission. The following is a list of the peer review reports and author responses from that submission.
Round 1
Reviewer 1 Report
This piece of research was a mammoth undertaking and great admiration for reaching that number of farmers. However, the standard of English and the liberal use of acronyms makes the article difficult to read. In addition the questions used appear to be too complicated for the average person, especially those with a low education as some were. I also have doubts about the 100% consent from farmers, which seems surprising when so many in each region were approached. Of course I recognise this could be a cultural difference. The questions were also generally positive statements, with only one negative statement, questionnaires should contain both to test the validity of the positive response.
Reviewer 2 Report
It's my great pleasure to review this paper. This is an interesting study. It takes a rural area in China area as a case, and explores the relationship model between farmers' ecological value cognition, environmental attitude and pro-environmental behavior. Furthermore, it discusses the mediating effect of environmental attitude and moderating effect of place attachment.
But there are still some problems that can be improved in the paper. First, the theoretical basis of the paper is relatively weak and needs to be further strengthened. The paper talks about the theory of planned behavior in line 100. The subsequent discussion lacks necessary explanation for this theory, which related fields the theory has made progress in, and how applicable the theory is to this study. Line 102 mentions that " Farmers are not only rational economic man, but also rational ecologic man ", which also lacks necessary explanations. Second, the literature review of the paper needs to be further improved. The author needs to supplement more relevant literature. And the summary is not detailed enough at present. At the same time, the research significance and value of this paper are not highlighted enough and need to be improved. Third, the method part of the paper also needs to be modified. The paper uses structural equation model, but the methods and steps are not accurate, for example, the data analysis process lacks confirmatory factor analysis. 4. After the analysis of the results, the paper also needs to explain the results in combination with the actual situation of the case and the collected materials in Jinan China. It can better explain the rationality of the model. 5. There are other problems in the paper. For example, "Figure 2" appears in line 131, but no relevant figure is found; Line 191 talks about 430.65 million people, which is not true; 33.3% of the people in Table 1 in row 206 have undergraduate 's degrees, which seems to be unrealistic.